# Novel Nitric Oxide Donor Dinitroazetidine-Coumarin Hybrids as Potent Anti-Intrahepatic Cholangiocarcinoma Agents

**DOI:** 10.3390/molecules27134021

**Published:** 2022-06-22

**Authors:** Zhihui Yu, Mengru Li, Shiqi Guo, Weijie Wang, Feng Qu, Yulei Ma, Hongrui Liu, Ying Chen

**Affiliations:** 1Department of Medicinal Chemistry, School of Pharmacy, Fudan University, Shanghai 201203, China; 18211030008@fudan.edu.cn (Z.Y.); 19211030083@fudan.edu.cn (S.G.); 20211030011@fudan.edu.cn (W.W.); 21211030007@m.fudan.edu.cn (F.Q.); 2Department of Pharmacology, School of Pharmacy, Fudan University, Shanghai 201203, China; 20211030050@fudan.edu.cn (M.L.); 19211030086@fudn.edu.cn (Y.M.)

**Keywords:** intrahepatic cholangiocarcinoma, NO donor, dinitroazetidine-coumarin hybrids, antitumor

## Abstract

Intrahepatic cholangiocarcinoma (iCC) is a serious liver cancer threatening human health. However, there are a few chemotherapeutic drugs for the treatment of iCC in the clinic. It is extremely urgent to develop new drugs for iCC. In this study, twenty dinitroazetidine and coumarin hybrids were synthesized and evaluated anti-iCC bioactivity as a new type of nitric oxide (NO) donors. Among them, compounds **2–5** and **21** showed a higher antiproliferative activity against RBE cell lines (human intrahepatic cholangiocarcinoma cell lines) and low cytotoxicity in nontumor cells (HOSEpiC and T29). The preliminary study of pharmacology mechanism indicated that compounds **2–5** and **21** could release effective concentration of NO in RBE cell lines, which leaded to inhibit the proliferation of RBE cell lines. The research results revealed that compound **3** inhibited the proliferation of RBE cell lines by inducing apoptosis and arresting cell cycle at G_2_/M phase. Additionally, compound **3** had acceptable metabolic stability. Therefore, compound **3** was merited to further explore for developing a desirable NO donor lead with anti-iCC activity.

## 1. Introduction

Cholangiocarcinoma (CCA) is a highly heterogeneous malignant tumor of the biliary tract that originates from the epithelial cells of the biliary duct and can occur anywhere in the biliary tree. Based on the anatomical location, CCA has three subtypes: intrahepatic, perihilar, and extrahepatic cholangiocarcinoma [1]. Among them, intrahepatic cholangiocarcinoma (iCC) is a primary liver cancer with high malignancy degree, difficult treatment, and poor prognosis, the incidence of which is second to hepatocellular carcinoma, accounting for about 10–15% of all primary liver cancer [2]. In recent decades, the incidence of iCC has been on the rise in all regions of the world [3].

Currently, therapies for iCC include surgical resection, ablation treatment, targeted treatment, and immunotherapy. Surgical resection has been a cornerstone in the management of iCC, which is an effective treatment for patients with early intrahepatic cholangiocarcinoma to achieve long-term survival. For patients with primary or recurrent iCC, percutaneous radiofrequency ablation (RFA) and percutaneous microwave ablation can be used [4]. The recurrence rate for iCC after surgical resection is as high as 40–80%, prompting a much greater need to develop and support strategies for adjuvant chemotherapeutic and targeted-agent therapeutics. National Comprehensive Cancer Network (NCCN) and Chinese Society of Clinical Oncology (CSCO) guidelines suggest that gemcitabine combining cisplatin (GP) and gemcitabine combining diageo (GS) are used as first-line standard chemotherapy drugs in the clinic [5]. At present, the results of other large-sample clinical trials of adjuvant chemotherapy for iCC are still lacking. For a subgroup of patients with known genetic alterations, such as *FGFR2* fusions or *IDH1* mutations, *FGFR2* inhibitor pemigatinib and *IDH1* inhibitor ivosidenib, as new drugs, have recently been approved as efficient subsequent treatment options for patients failing the first line of systemic chemotherapies [6]. However, the efficacy of these inhibitors has been shown to be short-lived due to acquired resistance [7]. Moreover, immune checkpoint inhibitors (ICIs) targeting the programmed cell death 1 (PD-1), the cytotoxic T-lymphocyte associated antigen 4 (CTLA-4), and other immunotherapy approaches have become standards of care for many cancers. They have demonstrated unprecedented efficacy, whereas the role for immunotherapy in iCC remains to be established [2,8,9]. Then, the development of small molecule chemotherapeutic drugs against iCC is still a critical unmet medical challenge.

Coumarin (2*H*-1-benzopyran-2-one) is common in nature, and its derivatives exhibit a fascinating array of pharmacological properties such as antibacterial, antifungal, antimalarial, and anticancer activities [10]. Coumarin derivatives can trigger various anticancer mechanisms, such as cell cycle arresting, apoptosis induction, and kinase signal pathway inhibition in different cancer cell lines. In our previous research, several furoxan-coumarin hybrids were synthesized and showed good proliferation inhibition activities in various tumor cell lines [11,12]. However, these furoxan derivatives have poor metabolic stability and unsatisfactory druggability. As we known, RRx-001, a dinitroazetidine-type NO donor, was able to release NO slowly and persistently in hypoxic tumor tissues to kill multiple tumors and had a relatively good metabolic stability. Currently, RRx-001 is undergoing to study in clinical phase III trials for the treatment of small cell lung cancer and might be a first nitric oxide donor antitumor drug. Therefore, in this study, we designed a class of new NO donor compounds by coupling dinitroazetidine moiety of RRx-001 and coumarin scaffold through amide-bond and aliphatic carbon-chain linkers, aiming to obtain a new lead compound with anti-iCC activity (Figure 1). Herein, the novel dinitroazetidine-coumarin hybrids were prepared and evaluated the antiproliferative activities against RBE cell lines and the cytotoxicity in two nontumor cells (HOSEpiC and T29). Meanwhile, NO releasing levels of target compounds, the apoptotic pathway, and cell cycle arrest in the tested cancer cell lines were also studied in this work.

## 2. Results and Discussion

### 2.1. Chemistry

As shown in Figure 1, the Mannich reaction of nitromethane, paraformaldehyde, and *tert*-Butylamine obtained intermediate **1a**, which was hydrolyzed with 10% hydrochloric acid to generate **1b**, further underwent the Mitsunobu reaction to get **1c** in the present of diisopropyl azodicarboxylate (DIAD) and triphenylphosphine (Ph_3_P). Nitrification reaction of **1c** with the NaNO_2_ and K_3_Fe(CN)_6_ of NaOH aqueous solution produced dinitroazetidine derivative **1d**, which reacted with acetic anhydride using BF_3_·Et_2_O as a catalyst to form amide **1e**. Finally, the key intermediate **1** was synthesized via the hydrolyzation of **1e** forming intermediate **1f** in the present of 10% hydrochloric acid and NaHCO_3_ aqueous solution, respectively.

The target compounds **2–21** were synthesized from coumarin derivatives and dinitroazetidine **1** using amide and carbon-chain as linkers, respectively. As depicted in Figure 2, under the catalysis of NaH, the nucleophilic substitution of ethyl acetoacetate with various commercially available bromides **2–5a** formed intermediates **2–5b**. At room temperature, the derivates **2–5c** bearing different substituents at 3-position of coumarin were prepared via the cyclization reaction of intermediates **2–5b** with resorcinol in 70–75% sulfuric acid. Moreover, 7-hydroxy in the compounds **2–5c** and **6a** were alkylated with 2-chloroethanol to get **2–5d** and **6b**. With the catalysis of NaH, the nucleophilic substitution of *tert*-butyl bromoacetate with various coumarin derivatives **2–5d** and **6b** formed intermediates **2–5e** and **6c**, from which removed the *tert*-butyl to obtain **2–5f** and **6d**. Then, in the presence of 2-(7-azabenzotriazol-1-yl)-*N*, *N*, *N*′, *N*′-tetramethyluronium hexafluorophosphate (HATU) and *N*, *N*-diisopropylethylamine (DIPEA), compounds **2–5f** and **6d** with carboxyl acid side chain were condensed with intermediate **1** to synthesize the amide linker target compounds **2–6**. The aliphatic carbon linker type target compounds **7–21** were synthesized via the etherification reaction of 7-hydroxycoumarin derivatives **2–5c** and **6a** with dibromide to produce the monobromide **7–21a**, and the nucleophilic substitution of **7–21a** with intermediate **1** in the present of K_2_CO_3_ and NaI.

### 2.2. In Vitro Antiproliferation Activities

As Figure 2 and Table 1 show, twenty target compounds **2–21** were screened for cytotoxicity at the concentration of 10 μM against RBE cell lines and two nontumorigenic cell lines (HOSEpiC and T29) with RRx-001, paclitaxel (PTX) and doxorubicin (DOX) as references using the MTT assays. The results showed that five compounds **2–5** and **21** displayed more than 50% antiproliferation activity in RBE cell lines (Figure 2a). Subsequently, they were further evaluated to figure out the values of IC_50_. As Table 1 described, four amide bond linker compounds (**2–5**) bearing 4- trifluoromethyl-benzyl, 4-cyanobenzyl, 4-fluorobenzyl and benzyl substituted at 3-position of coumarin and 4C-chain linker compound **21** without group at 3-position of coumarin exhibited stronger antiproliferation effects in RBE cell lines with the values of IC_50_ ranging from 0.71 to 1.11 μM compared to RRx-001 with the 2.00 μM of IC_50_. Moreover, we evaluated the toxicity of the compounds **2–21** in HOSEpiC and T29. As Figure 2b,c shown, most of the target compounds did not exhibit significant cytotoxicity with cell viability higher than 95% in two nontumor cell lines, which indicated these newly synthesized dinitroazetidine-coumarin hybrids had a good safety.

The antiproliferation activities of individual compound to tumor cells were determined by the MTT assay.

### 2.3. Nitric Oxide Releasing in RBE Cell Lines

As we known, anticancer activity of RRx-001 is relative to NO releasing level of its dinitroazetidine moiety [13]. Considering that compounds **2–21** were synthesized through the combination of dinitroazetidine moiety from RRx-001 and coumarin derivatives and that they showed better inhibitory activity in RBE cell lines compared with RRx-001, we then explored whether these compounds could also release NO comparable to RRx-001 in RBE cell lines. The nitric oxide release of RRx-001 and active compounds **2–5** and **21** in RBE cell lines was determined using the fluorescent probe DAF-FM DA. As Figure 3 presented, compared to RRx-001, the exposure of RBE cells to compounds **2–5** and **21** with the concentration of 2 μM for 2.5 h led to approximately same level of fluorescence intensity. This result implicated that these hybrids can release a relevant concentration of NO in RBE cells, which is closely related to their good inhibitory activities in RBE cell lines. Among them, compound **3** had the highest NO release concentration.

### 2.4. Compound ***3*** Blocked Cell Cycle and Induced Apoptosis

The cell cycle of eukaryotic cells is the basic process of cell life action, in which DNA synthesis and cell division are the two main events [14]. Many reports showed that coumarin derivatives inhibited tumor cell proliferation through cell cycle arrest and inducing apoptosis [10,12,14]. In our previous study, furoxan-coumarin hybrids arrested A2780 cell cycle in G_2_/M phase [12]. Therefore, we performed the cell cycle arrest assay of these dinitroazetidine-coumarin hybrids in RBE cell lines. As Figure 4a shown, DMSO as a control, after treating RBE cell lines using compound **3** with the concentration of 1 μM, the mean percentage of cells in the G_2_/M phase increased from 18 to 23% and the percentages of cells in S and G_0_/G_1_ phase decreased concomitantly. This result implied that compound **3** was able to arrest the cell cycle at G_2_/M phase. Additionally, Western blotting analysis displayed that compound **3** apparently downregulated the expression of the antiapoptotic proteins PARP and Caspase-3 in a dose-dependent manner and cell cycle protein Cyclin B_1_, which is involved in the G_2_/M phase regulation [15] (Figure 4b). These results deduced that the antiproliferative ability of compound **3** might involve in the mechanism of arresting cell cycle in G_2_/M phase and inducing apoptotic pathway.

### 2.5. Metabolic Stability in Liver Microsomes

In this work, one of the purposes was to obtain new nitric oxide donor compounds with improved metabolic stability compared to furoxan derivatives. Therefore, we evaluated the metabolic stability of compound **3** and furoxan-coumarin hybrid **CY-14S-4A83** in liver microsomes of human, rat, and mouse. The results showed that the newly synthesized dinitroazetidine-coumarin hybrid **3** has an obviously improved metabolic stability with 3.01, 9.44, and 6.48 of MF% (metabolic bioavailability) in the human, rat, and mouse, which were better than that furoxan-coumarin derivatives with lower than 0.5 of MF% in the same liver microsomes (Table 2).

Compounds **CY-14S-4A83** and **3** (0.1 μM) were incubated with liver microsomes of different species (0.33 mg/mL) at 37 °C; then, the samples were analyzed by LC-MS/MS. 

## 3. Materials and Methods

### 3.1. General Information

Chemicals were purchased from the following suppliers: Sinopharm, Adamas, Merck, and Sigma Aldrich. Solvents were dried before use, if required. Air- and moisture-sensitive reactions were carried out under nitrogen atmosphere. Room temperature (r.t.) refers to 20–25 °C. The progress of a reaction was monitored by thin layer chromatography (TLC) using precoated TLC sheets purchased from Sinopharm. Detected spots were observed under UV light at λ 254 and 365 nm. Melting points were measured on a SGW X-4 microscopy melting point apparatus without correction. ^1^H-NMR and ^13^C-NMR spectral data (Appendix A) were recorded with a Bruker DRX 600MHz spectrometer, both at 303 K using TMS as an internal standard. All chemical shifts are reported in ppm (*δ*) and coupling constants (*J*) are in hertz (Hz). Mass spectra were recorded on Agilent Technologies 1260 infinity LC/MS instrument. The chromatograms were conducted on silica gel (100–200 and 300–400 mesh) and visualized under UV light at λ 254 and 365 nm.

### 3.2. Synthesis

***N-tert-butyl-5-hydroxymethyl-5-nitro-1,3-oxazine*** (**1a**): To aqueous solution of paraformaldehyde (24 g, 0.8 mol) and 40%NaOH (600 μL) in 120 mL of distilled water, nitromethane (10.5 mL, 0.195 mol) was added dropwise over 1 h at 40 °C. The reaction mixture was heated to 60 °C and stirred for 1 h. Then, the solution of tert-butylamine (20.3 mL, 0.262 mol) in distilled water (36 mL) was added dropwise slowly. The mixture was stirred for another 4 h, cooled to room temperature, and stirred for 1 h again. The precipitate was collected by vacuum filtration at room temperature, washed with distilled water, and vacuum freeze-dried to give **1a** (yellow solid, 34 g, 78.8% yield). ESI-MS *m*/*z* 231.1 [M + H]^+^.

***N-****(**tert-butylamino**)**methyl-2-nitro-1,3-propandiolhydrochloride**
*(**1b**): To a solution of concentrated hydrochloric acid (6.71 mL, 81 mmol) in methanol (62.5 mL), **1a** (17.4 g, 80 mmol) was added. The reaction solution was refluxed for 20 h. The solvent was removed through vacuum evaporation and the residue was dissolved in isopropyl alcohol (25 mL). The solution was recrystallized below 0 °C and the precipitate was filtered, washed with isopropanol, and vacuum freeze-dried to give **1b** (white solid, 10.4 g, 54% yield). ESI-MS *m*/*z* 207.0 [M + H]^+^.

***N-tert-butyl-3-hydroxymethyl-3-nitroazetidine hydrochloride*** (**1c**): To a solution of DIAD (5.15 mL, 25.98 mmol) and **1b** (5.0 g, 20.60 mmol) in butanone (40 mL), Ph_3_P (29.89 g, 0.132 mol) in butanone was added dropwise over 1 h at 50 °C. The reaction mixture was stirred at 50 °C for 4 h, filtered, washed with cold butanone (30 mL), and vacuum freeze-dried to give **1c** (white solid,3.0 g, 65% yield, m.p. 161.8–163.5 °C). ESI-MS *m*/*z* 189.0 [M + H]^+^.

***N-tert-butyl-3,3-dinitroazetidine*** (**1d**): To a solution of **1c** (1.35 g, 6 mmol) in distilled water (6 mL), NaOH aqueous solution (3 mL, 717 mg, 17.9 mmol) was added and was stirred for 3 h at room temperature. After cooling to 8 °C, cold NaNO_2_ solution (4.5 mL, 1.65 g, 23.9 mmol) and K_3_Fe(CN)_6_ (197 mg, 6 mmol) in distilled water were added slowly. Then, Na_2_S_2_O_8_ (1.78 g, 7.5 mmol) was added. The yellow solution was stirred for another 1 h at room temperature and extracted with dichloromethane (150 mL). The organic layer was dried with MgSO_4_ and the solvent was removed via vacuum evaporation to give **1d** (yellow liquid, 858 mg, 70.5% yield). ESI-MS *m*/*z* 203.9 [M + H]^+^.

***N-acetyl-3,3-dintroazetidine*** (**1e**): Compound **1d** (1.0 g, 4.92 mmol) and acetic anhydride (1.8 mL, 19.98 mmol) were slowly added into the reaction system, followed by injecting boron trifluoride etherate (1.31 mL, 0.492 mmol) using syringe. The mixture was reacted at 115–125 °C for 3 h under nitrogen atmosphere. Excess acetic anhydride was removed by vacuum distillation. The residue was recrystallized in chloroform to give **1e** (white crystal, 312 mg, 33.6% yield, m.p. 112.3–114.5 °C). ESI-MS *m*/*z* 189.9 [M + H]^+^. ^1^H NMR (600 MHz, DMSO) δ 5.07 (s, 2H), 4.74 (s, 2H), 1.87 (s, 3H).

***3,3-Dintroazetidine hydrochloride*** (**1f**): To a solution of **1e** (150 mg, 0.817 mmol) in distilled water (20 mL), 10% hydrochloric acid (1.5 mL) was added dropwise. The solution was stirred and refluxed for 4 h. The solvent was removed by vacuum evaporation to give **1f** (white solid, 100 mg, 66.7% yield). ESI-MS *m*/*z* 148.1 [M + H]^+^.

***3,3-Dintroazetidine*** (**1**): The solution of **1f** (1.0 g, 5.45 mmol) in distilled water (32 mL) was heated to 50 °C, and 10% NaHCO_3_ was slowly added dropwise until pH-9. The mixture was extracted with chloroform (3 × 20 mL), and the organic layer was dried with MgSO_4_. The solvent was removed under lowered pressure to give intermediate **1** (light yellow oil, 400 mg, 50% yield). ESI-MS *m*/*z* 148.1 [M + H]^+^. ^1^H NMR (600 MHz, DMSO) δ 4.34 (s, 4H).

**General Procedure for the Preparation of 2–5b.** In an ice bath, a reaction solution of 60% NaH (680 mg, 17 mmol) in dry THF (40 mL) was stirred for 10 min. Ethyl acetoacetate (2.2 mL, 17 mmol) was added dropwise. After stirring the reaction for 30 min, starting materials **2–5a** (15.3 mmol) were added and continued reaction at room temperature in the TLC monitor. After the reaction finished, the solids were filtered out, most of the solvents were removed, and water (100 mL) was added. The mixture was extracted with ethyl acetate, washed with saturated salt, dried over anhydrous sodium sulfate, and evaporated *in vacuo* to get compounds **2–5b**. These products were directly used as the next reactants without any further purification.

***Ethyl 3-oxo-2-****(**4-**(**trifluoromethyl**)**benzyl**)**butanoate**
*(**2b**)**.** The title compound was obtained starting from **2a** and ethyl acetoacetate. Analytical data for **2b** (a yellow oily substance, 4.2 g, 95% yield): ESI-MS *m/z* 289.2 [M + H]^+^.

***Ethyl 2-****(**4-cyanobenzyl**)**-3-oxobutanoate**
*(**3b**)**.** The title compound was obtained starting from **3a** and ethyl acetoacetate. Analytical data for **3b** (a yellow oily substance, 3.7 g, 98% yield): ESI-MS *m/z* 245.9 [M + H]^+^.

***Ethyl 2-****(**4-fluorobenzyl**)**-3-oxobutanoate**
*(**4b**)**.** The title compound was obtained starting from **4a** and ethyl acetoacetate. Analytical data for **4b** (a yellow oily substance, 3.6 g, 97% yield): ESI-MS *m/z* 260.9 [M + Na]^+^.

***Ethyl 2-benzyl-3-oxobutanoate*** (**5b**)**.** The title compound was obtained starting from **5a** and ethyl acetoacetate. Analytical data for **5b** (a yellow oily substance, 3.3 g, 98% yield): ESI-MS *m/z* 221.0 [M + H]^+^.

**General Procedure for the Preparation of 2–5c.** Resorcinol (1.35 g, 12.250 mmol) was added to a reaction flask containing **2–5b** and 70% H_2_SO_4_ (30 mL) in the ice bath. After 30 min, the ice bath was removed and the reaction continued at room temperature. After the reaction finished, the solution was slowly added into ice water (300 mL) and stirred for 30 min to precipitate a large number of solids. After extraction, filtration, and drying, compounds **2–5c** were obtained.

***7-Hydroxy-4-methyl-3-****(**4-**(**trifluoromethyl**)**benzyl**)**-2H-chromen-2-one*** (**2c**)**.** The title compound was obtained starting from **2b** and resorcinol. Analytical data for **2c** (white solid, 3.4 g 82% yield): ESI-MS *m*/*z* 335.2 [M + H]^+^.

***4-****((**7-Hydroxy-4-methyl-2-oxo-2H-chromen-3-yl**)**methyl**)**benzonitrile**
*(**3c**)**.** The title compound was obtained starting from **3b** and resorcinol. Analytical data for **3c** (white solid, 3.0 g, 83% yield): ESI-MS *m/z* 292.3 [M + H]^+^.

***3-****(**4-Fluorobenzyl**)**-7-hydroxy-4-methyl-2H-chromen-2-one**
*(**4c**)**.** The title compound was obtained starting from **4b** and resorcinol. Analytical data for **4c** (yellow solid, 3.2 g, 91% yield): ESI-MS *m/z* 284.9 [M + H]^+^.

***3-Benzyl-7-hydroxy-4-methyl-2H-chromen-2-one*** (**5c**)**.** The title compound was obtained starting from **5b** and resorcinol. Analytical data for **5c** (yellow solid, 2.7 g, 83% yield): ESI-MS *m/z* 266.9 [M + H]^+^.

**General Procedure for the Preparation of 2–5d and 6b.** To a stirred solution of **2–5c** and **6a** (3.517 mmol) in DMF (15 mL) at room temperature, corresponding halo alcohol (10.553 mmol), NaI (1.05 mmol), and K_2_CO_3_ (10.553 mmol) were added. The mixture was refluxed for 2–5 h and then poured into ice water (50 mL). After filtration, the residue was washed with water (3 × 10 mL) and dried to obtain **2–5d** and **6b**.

***7-****(**2-hydroxyethoxy**)**-4-methyl-3-**(**4-**(**trifluoromethyl**)**benzyl**)**-2H-chromen-2-one**
*(**2d**)**.** The title compound was obtained starting from **2c** and ethylene chlorohydrin. Analytical data for **2d** (white solid, 1.1 g, 81% yield, m.p. 104–106 °C): ESI-MS *m/z* 379.1 [M + H]^+^.

***4-****((**7-**(**2-hydroxyethoxy**)**-4-methyl-2-oxo-2H-chromen-3-yl**)**methyl**)**benzonitrile**
*(**3d**)**.** The title compound was obtained starting from **3c** and ethylene chlorohydrin. Analytical data for **3d** (white solid, 978 mg, 83% yield, m.p. 80–82 °C): ESI-MS *m/z* 336.3 [M + H]^+^.

***3-****(**4-fluorobenzyl**)**-7-**(**2-hydroxyethoxy**)**-4-methyl-2H-chromen-2-one**
*(**4d**)**.** The title compound was obtained starting from **4c** and ethylene chlorohydrin. Analytical data for **4d** (white solid, 992 mg, 81% yield): ESI-MS *m/z* 329.1 [M + H]^+^.

***3-benzyl-7-****(**2-hydroxyethoxy**)**-4-methyl-2H-chromen-2-one**
*(**5d**)**.** The title compound was obtained starting from **5c** and ethylene chlorohydrin. Analytical data for **5d** (white solid, 861 mg, 81% yield): ESI-MS *m/z* 311.1 [M + H]^+^.

***7-****(**2-Hydroxyethoxy**)**-4-methyl-2H-chromen-2-one*** (**6b**)**.** The title compound was obtained starting from **6a** and ethylene chlorohydrin. Analytical data for **6b** (white solid, 619 mg, 81% yield): ESI-MS *m/z* 221.1 [M + H]^+^.

**General Procedure for the Preparation of 2–5e and 6c.** To a stirred solution of **2–5d** and **6b** (4.54 mmol) in anhydrous DMF (15 mL) at 0 °C, NaH (60%, 363 mg, 9.08 mmol) was added. The mixture was stirred for 20 min and *t*-butylbromoacetate (1.34 mL, 9.08 mmol) was added. The reaction mixture was warmed to room temperature and stirred for 1.5 h, and then poured into a saturated aqueous solution of NH_4_Cl (10 mL) and extracted with ethyl acetate (3 × 30 mL). The combined extracts were washed with brine, dried over Na_2_SO_4_, filtered, and evaporated *in vacuo*. The residue was purified by column chromatography on silica gel to give compound **2–5e** and **6c**.

***T-butyl 2-****(**2-**((**4-methyl-2-oxo-3-**(**4-**(**trifluoromethyl**)**benzyl**)**-2H-chromen-7-yl**)**oxy**) **ethoxy**)**acetate*** (**2e**): The title compound was obtained starting from **2d**. Analytical data for **2e** (colorless liquid, 446 mg, 25% yield): ESI-MS *m*/*z* 436.8 [M + H]^+^; ^1^H NMR (600 MHz, DMSO) δ 7.67 (d, *J* = 7.7 Hz, 2H), 7.57 (d, *J* = 7.7 Hz, 2H), 6.92 (d, *J* = 8.1 Hz, 1H), 6.47 (d, *J* = 13.2 Hz, 2H), 4.23 (s, 2H), 3.96 (s, 2H), 3.93 (s, 2H), 3.70 (d, *J* = 4.5 Hz, 2H), 2.05 (s, 3H), 1.43 (s, 9H).

***Tert-butyl 2-****(**2-**((**3-**(**4-cyanobenzyl**)**-4-methyl-2-oxo-2H-chromen-7-yl**)**oxy**)**ethoxy**) **acetate*** (**3e**): The title compound was obtained starting from **3d**. Analytical data for **3e** (white acicular crystal, 239 mg, 13.4% yield, m.p. 90.5–92.4 °C): ESI-MS *m*/*z* 393.9 [M + H]^+^; ^1^H NMR (600 MHz, DMSO) δ 7.76 (t, *J* = 13.1 Hz, 3H), 7.44 (d, *J* = 7.5 Hz, 2H), 7.01 (dd, *J* = 20.5, 11.6 Hz, 2H), 4.24 (s, 2H), 4.07 (s, 2H), 4.04 (s, 2H), 3.84 (s, 2H), 2.44 (s, 3H), 1.43 (s, 9H).

***Tert-butyl 2-****(**2-**((**3-**(**4-fluorobenzyl**)**-4-methyl-2-oxo-2H-chromen-7-yl**)**oxy**)**ethoxy**) **acetate*** (**4e**): The title compound was obtained starting from **4d**. Analytical data for **4e** (yellow liquid, 301 mg, 14.8% yield): ESI-MS *m*/*z* 442.9 [M + H]^+^, ^1^H NMR (600 MHz, DMSO) δ 7.35 (dd, *J* = 8.2, 5.8 Hz, 2H), 7.12 (t, *J* = 8.8 Hz, 2H), 6.89 (d, *J* = 8.2 Hz, 1H), 6.49–6.42 (m, 2H), 4.23 (s, 2H), 3.95 (t, *J* = 4.9 Hz, 2H), 3.81 (s, 2H), 3.70 (dd, *J* = 10.3, 5.2 Hz, 2H), 2.05 (s, 3H), 1.43 (s, 9H).

***Tert-butyl 2-****(**2-**((**3-benzyl-4-methyl-2-oxo-2H-chromen-7-yl**)**oxy**)**ethoxy**)**acetate**
*(**5e**)*:* The title compound was obtained starting from **5d**. Analytical data for **5e** (yellow liquid, 442 mg, 23% yield): ESI-MS *m*/*z* 424.9 [M + H]^+^; ^1^H NMR (600 MHz, DMSO) δ 7.32 (t, *J* = 5.4 Hz, 2H), 7.29 (d, *J* = 7.8 Hz, 1H), 7.19 (t, *J* = 6.9 Hz, 1H), 6.89 (d, *J* = 8.2 Hz, 1H), 6.46 (dd, *J* = 12.7, 4.3 Hz, 2H), 4.23 (s, 2H), 3.95 (t, *J* = 4.9 Hz, 2H), 3.83 (s, 2H), 3.70 (dd, *J* = 10.2, 5.1 Hz, 2H), 2.05 (s, 3H), 1.32 (s, 9H).

***Tert-butyl 2-****(**2-**((**4-methyl-2-oxo-2H-chromen-7-yl**)**oxy**)**ethoxy**)**acetate**
*(**6c**)*:* The title compound was obtained starting from **6b**. Analytical data for **6c** (white solid, 240 mg, 19% yield, m.p. 79.0–80.9 °C): ESI-MS *m*/*z* 279.1 [M + H]^+^; ^1^H NMR (600 MHz, DMSO) δ 7.69 (d, *J* = 8.7 Hz, 1H), 7.04–6.95 (m, 2H), 6.22 (s, 1H), 4.24 (s, 2H), 4.07 (s, 2H), 3.84 (s, 2H), 2.40 (s, 3H), 1.43 (s, 9H).

**General Procedure for the Preparation of 2–5f and 6d.** To a stirred solution of **2–5e** and **6c** (50 mg) in DCM (5 mL) at 0 °C was added TFA (100 μL). The reaction mixture was warmed to room temperature and stirred for 2 h. After the reaction finished, the solvent and unreacted TFA were evaporated in vacuo to get compounds **2–5f** and **6d**.

***2-****(**2-**((**4-methyl-2-oxo-3-**(**4-**(**trifluoromethyl**)**benzyl**)**-2H-chromen-7-yl**)**oxy**)**ethoxy**) **acetic acid**
*(**2f**): The title compound was obtained starting from **2e**. Analytical data for **2f** (brown solid, 48 mg, 99% yield): ESI-MS *m*/*z* 436.8 [M + H]^+^.

***2-****(**2-**((**3-**(**4-cyanobenzyl**)**-4-methyl-2-oxo-2H-chromen-7-yl**)**oxy**)**ethoxy**)**acetic acid*** (**3f**)*:* The title compound was obtained starting from **3e**. Analytical data for **3f** (brown solid, 43 mg, 100% yield): ESI-MS *m*/*z* 393.8 [M + H]^+^.

***2-****(**2-**((**3-**(**4-fluorobenzyl**)**-4-methyl-2-oxo-2H-chromen-7-yl**)**oxy**)**ethoxy**)**acetic acid**
*(**4f**): The title compound was obtained starting from **4e**. Analytical data for **4f** (brown solid, 39 mg, 88.8% yield): ESI-MS *m*/*z* 386.9 [M + H]^+^.

***2-****(**2-**((**3-**(**4-fluorobenzyl**)**-4-methyl-2-oxo-2H-chromen-7-yl**)**oxy**)**ethoxy**)**acetic acid**
*(**5f**): The title compound was obtained starting from **5e**. Analytical data for **5f** (yellow solid, 43 mg, 99% yield): ESI-MS *m*/*z* 368.9 [M + H]^+^.

***2-****(**2-**((**4-methyl-2-oxo-2H-chromen-7-yl**)**oxy**)**ethoxy**)**acetic acid**
*(**6d**): The title compound was obtained starting from **6c**. Analytical data for **6d** (brown solid, 41 mg, 98% yield): ESI-MS *m*/*z* 278.9 [M + H]^+^.

**General Procedure for the Preparation of 2–6.** Substituted carboxylic acid **2–5f** and **6d** (0.2031 mmol) were dissolved in DCM (10 mL) and stirred for 30 min at room temperature. DIPEA (71 μL, 0.4062 mmol) and HATU (154 mg, 0.4062 mmol) were added to the solution and stirred for 40 min at room temperature. After being added intermediate **1** (60 mg, 0.4062 mmol), the mixture was further stirred at room temperature for another 4 h, then water was added. The mixture was extracted with ethyl acetate, washed with saturated salt, and dried over anhydrous Na_2_SO_4_ to obtain crude product, which was purified by column chromatography on silica gel to give compounds **2–6**.

***7-****(**2-**(**2-**(**3,3-Dinitroazetidin-1-yl**)**-2-oxoethoxy**)**ethoxy**)**-4-methyl-3-**(**4-**(**trifluoromethyl**)**benzyl**)**-2H-chromen-2-one**
*(**2**): The title compound was obtained starting from **2f** and 3,3-dinitroazetidine. Analytical data for **2** (yellow solid, 34 mg, 30.7% yield, m.p. 65.3–67.0 °C): ESI-MS *m/z* 565.7 [M + H]^+^; ^1^H NMR (600 MHz, DMSO) δ 7.67 (d, *J* = 7.9 Hz, 2H), 7.55 (d, *J* = 7.9 Hz, 2H), 6.97 (d, *J* = 8.8 Hz, 1H), 6.52 (s, 2H), 5.13 (s, 2H), 4.91 (s, 2H), 4.37 (s, 2H), 3.97 (t, *J* = 4.7 Hz, 2H), 3.94 (s, 2H), 3.72 (t, *J* = 9.8, 4.8 Hz, 2H), 2.07 (s, 3H). ^13^C NMR (151 MHz, DMSO) δ 168.98, 167.58, 158.96, 154.28, 146.09, 143.92, 128.57, 127.09, 125.08, 125.06, 124.41, 107.41, 107.09, 105.59, 99.92, 69.41, 66.15, 60.43, 59.37, 34.52, 21.91.

***4-****((**7-**(**2-**(**2-**(**3,3-Dinitroazetidin-1-yl**)**-2-oxoethoxy**)**ethoxy**)**-4-methyl-2-oxo-2H-chromen-3-yl**)**methyl**)**benzonitrile*** (**3**): The title compound was obtained starting from **3f** and 3,3-dinitroazetidine. Analytical data for **3** (yellow solid, 30 mg, 28% yield, m.p. 74.9–76.3 °C): ESI-MS *m*/*z* 522.7 [M + H]^+^; ^1^H NMR (600 MHz, DMSO) δ 7.82–7.70 (m, 3H), 7.44 (d, *J* = 6.0 Hz, 2H), 7.07–6.97 (m, 2H), 5.15 (s, 2H), 4.80 (s, 2H), 4.28 (s, 2H), 4.18 (s, 2H), 4.05 (s, 2H), 3.83 (s, 2H), 2.44 (s, 3H). ^13^C NMR (151 MHz, DMSO) δ 170.16, 160.98, 160.79, 153.28, 149.02, 145.27, 132.18, 128.98, 126.66, 119.82, 118.74, 113.42, 112.35, 108.80, 107.68, 100.90, 69.54, 68.93, 67.36, 59.56, 56.77, 32.22, 15.10.

***7-****(**2-**(**2-**(**3,3-Dinitroazetidin-1-yl**)**-2-oxoethoxy**)**ethoxy**)**-3-**(**4-fluorobenzyl**)**-4-methyl-2H-chromen-2-one**
*(**4**): The title compound was obtained starting from **4f** and 3,3-dinitroazetidine. Analytical data for **4** (yellow solid, 37 mg, 35% yield, m.p. 70.0–71.9 °C): ESI-MS *m*/*z* 515.8 [M + H]^+^; ^1^H NMR (600 MHz, DMSO) δ 7.39–7.31 (m, 2H), 7.11 (t, *J* = 8.6 Hz, 2H), 6.94 (d, *J* = 8.0 Hz, 1H), 6.51 (d, *J* = 8.5 Hz, 2H), 5.12 (s, 2H), 4.89 (s, 2H), 4.36 (s, 2H), 3.97 (t, *J* = 4.6 Hz, 2H), 3.82 (s, 2H), 3.71 (t, *J* = 4.7 Hz, 2H), 2.07 (s, 3H). ^13^C NMR (151 MHz, DMSO) δ 168.92, 167.60, 166.89, 158.92, 154.32, 144.95, 134.90, 129.57, 128.66, 128.07, 124.49, 114.96, 114.82, 107.43, 107.10, 105.59, 99.94, 69.40, 66.14, 60.42, 59.37, 58.95, 56.91, 33.87, 21.68.

***3-Benzyl-7-****(**2-**(**2-**(**3,3-dinitroazetidin-1-yl**)**-2-oxoethoxy**)**ethoxy**)**-4-methyl-2H-chromen-2-one**
*(**5**): The title compound was obtained starting from **5f** and 3,3-dinitroazetidine. Analytical data for **5** (white solid, 30 mg, 30% yield, m.p. 82.5–84.5 °C): ESI-MS *m*/*z* 497.8 [M + H]^+^; ^1^H NMR (600 MHz, DMSO) δ 7.35–7.25 (m, 5H), 6.94 (d, *J* = 8.0 Hz, 1H), 6.51 (d, *J* = 7.8 Hz, 2H), 5.13 (s, 2H), 4.86 (s, 2H), 4.72 (s, 2H), 4.35 (s, 2H), 3.97 (t, *J* = 4.9 Hz, 2H), 3.84 (s, 2H), 3.71 (dd, *J* = 10.0, 5.0 Hz, 2H), 2.08 (s, 3H). ^13^C NMR (151 MHz, DMSO) δ 168.92, 167.63, 158.91, 154.36, 144.78, 138.83, 128.73, 128.23, 127.83, 125.81, 124.57, 107.45, 107.11, 105.61, 99.96, 69.40, 66.21, 60.47, 59.37, 34.70, 21.66.

***7-****(**2-**(**2-**(**3,3-dinitroazetidin-1-yl**)**-2-oxoethoxy**)**ethoxy**)**-4-methyl-2H-chromen-2-one**
*(**6**): The title compound was obtained starting from **5f** and 3,3-dinitroazetidine. Analytical data for **5** (white solid, 37 mg, 30% yield, m.p. 82.5–84.5 °C): ESI-MS *m*/*z* 408.1 [M + H]^+^; ^1^H NMR (600 MHz, DMSO) δ 7.70 (d, *J* = 8.8 Hz, 1H), 7.06–6.95 (m, 2H), 6.22 (s, 1H), 5.16 (s, 2H), 4.80 (s, 2H), 4.33–4.24 (m, 2H), 4.18 (s, 2H), 3.87–3.78 (m, 2H), 2.41 (s, 3H). ^13^C NMR (151 MHz, DMSO) δ 170.16, 161.26, 159.97, 154.57, 153.25, 126.35, 113.08, 112.22, 111.05, 107.68, 101.11, 69.54, 68.91, 67.37, 59.56, 56.77, 17.97.

**General Procedure for the Preparation of 7–21a.** Compounds **2–5c** and **6a** (500 mg, 1.76 mmol) were dissolved in CH_3_CN (15 mL) in a three-necked flask. Then K_2_CO_3_ (1.2 g, 8.8 mmol) and various dibromo alkane (8.8 mmol) were added to the reaction mixture, which was heated to 80 °C and stirred for 7 h. After finishing, the mixture was cooled to room temperature and filtered. The filtrate was concentrated *in vacuo*; then, DCM (20 mL) and H_2_O (10 mL) were added. The mixture was extracted with DCM, washed with saturated salt and dried over anhydrous Na_2_SO_4_ to obtain compounds **7–21a**.

***7-****(**2-Bromoethoxy**)**-4-methyl-3-**(**4-**(**trifluoromethyl**)**benzyl**)**-2H-chromen-2-one**
*(**7a**): The title compound was obtained starting from **2c** and 1,2-dibromoethane. Analytical data for **7a** (white solid, 658 mg, 85% yield): ESI-MS *m*/*z* 440.7 [M + H]^+^.

***7-****(**3-Bromopropoxy**)**-4-methyl-3-**(**4-**(**trifluoromethyl**)**benzyl**)**-2H-chromen-2-one**
*(**8a**): The title compound was obtained starting from **2c** and 1,3-dibromopropane. Analytical data for **8a** (white solid, 663 mg, 83% yield): ESI-MS *m*/*z* 454.7 [M + H]^+^.

***7-****(**4-Bromobutoxy**)**-4-methyl-3-**(**4-**(**trifluoromethyl**)**benzyl**)**-2H-chromen-2-one**
*(**9a**): The title compound was obtained starting from **2c** and 1,4-dibromobutane. Analytical data for **9a** (white solid, 750 mg, 91% yield): ESI-MS *m*/*z* 468.7 [M + H]^+^.

***4-****((**7-**(**2-Bromoethoxy**)**-4-methyl-2-oxo-2H-chromen-3-yl**)**methyl**)**benzonitrile**
*(**10a**): The title compound was obtained starting from **3c** and 1,2-dibromoethane. Analytical data for **10a** (white solid, 613 mg, 88% yield): ESI-MS *m*/*z* 397.0 [M + H]^+^.

***4-****((**7-**(**3-Bromopropoxy**)**-4-methyl-2-oxo-2H-chromen-3-yl**)**methyl**)**benzonitrile**
*(**11a**): The title compound was obtained starting from **3c** and 1,3-dibromopropane. Analytical data for **11a** (white solid, 434 mg, 83% yield): ESI-MS *m*/*z* 411.8 [M + H]^+^.

***4-****((**7-**(**4-Bromobutoxy**)**-4-methyl-2-oxo-2H-chromen-3-yl**)**methyl**)**benzonitrile**
*(**12a**): The title compound was obtained starting from **3c** and 1,4-dibromobutane. Analytical data for **12a** (yellow solid, 598 mg, 80% yield): ESI-MS *m*/*z* 425.9 [M + H]^+^.

***7-****(**2-Bromoethoxy**)**-3-**(**4-fluorobenzyl**)**-4-methyl-2H-chromen-2-one**
*(**13a**): The title compound was obtained starting from **4c** and 1,2-dibromoethane. Analytical data for **13a** (yellow solid, 665 mg, 97.3% yield): ESI-MS *m*/*z* 390.8 [M + H]^+^.

***7-****(**3-Bromopropoxy**)**-3-**(**4-fluorobenzyl**)**-4-methyl-2H-chromen-2-one**
*(**14a**): The title compound was obtained starting from **4c** and 1,3-dibromopropane. Analytical data for **14a** (white solid, 633 mg, 89% yield): ESI-MS *m*/*z* 404.8 [M + H]^+^.

***7-****(**4-Bromobutoxy**)**-3-**(**4-fluorobenzyl**)**-4-methyl-2H-chromen-2-one**
*(**15a**): The title compound was obtained starting from **4c** and 1,4-dibromobutane. Analytical data for **15a** (white solid, 625 mg, 77% yield): ESI-MS *m*/*z* 418.8 [M + H]^+^.

***3-Benzyl-7-****(**2-bromoethoxy**)**-4-methyl-2H-chromen-2-one*** (**16a**): The title compound was obtained starting from **5c** and 1,2-dibromoethane. Analytical data for **16a** (white solid, 478 mg, 81% yield): ESI-MS *m*/*z* 336.3 [M + H]^+^.

***3-Benzyl-7-****(**3-bromopropoxy**)**-4-methyl-2H-chromen-2-one**
*(**17a**): The title compound was obtained starting from **5c** and 1,3-dibromopropane. Analytical data for **17a** (white solid, 530 mg, 78% yield): ESI-MS *m*/*z* 387.0 [M + H]^+^.

***3-Benzyl-7-****(**4-bromobutoxy**)**-4-methyl-2H-chromen-2-one*** (**18a**): The title compound was obtained starting from **5c** and 1,4-dibromobutane. Analytical data for **18a** (white solid, 543 mg, 77% yield): ESI-MS *m*/*z* 402.4 [M + H]^+^.

***7-****(**2-Bromoethoxy**)**-4-methyl-2H-chromen-2-one**
*(**19a**): The title compound was obtained starting from **6a** and 1,2-dibromoethane. Analytical data for **19a** (white solid, 402 mg, 81% yield): ESI-MS *m*/*z* 283.0 [M + H]^+^.

***7-****(**3-Bromopropoxy**)**-4-methyl-2H-chromen-2-one**
*(**20a**): The title compound was obtained starting from **6a** and 1,3-dibromopropane. Analytical data for **20a** (white solid, 425 mg, 81% yield): ESI-MS *m*/*z* 298.9 [M + H]^+^.

***7-****(**4-Bromobutoxy**)**-4-methyl-2H-chromen-2-one**
*(**21a**): The title compound was obtained starting from **6a** and 1,4-dibromobutane. Analytical data for **21a** (white solid, 387 mg, 71% yield): ESI-MS *m*/*z* 310.8 [M + H]^+^.

**General Procedure for the Preparation of 7–21.** Compounds **7–21a** (0.34 mmol), DMAP (166 mg, 1.36 mmol) and 3,3-dinitroazetidine (150 mg, 1.02 mmol) were dissolved in DMF (10 mL) in a three-necked flask. This mixture was heated to 40–80 °C and stirred for 6–8 h. After finishing, the mixture was cooled to room temperature and added water. Then, the reaction mixture was extracted with ethyl acetate, washed with saturated salt and dried over anhydrous Na_2_SO_4_ to obtain crude product, which was purified by column chromatography on silica gel to give the target compounds **7–21**.

***7-****(**2-**(**3,3-Dinitroazetidin-1-yl**)**ethoxy**)**-4-methyl-3-**(**4-**(**trifluoromethyl**)**benzyl**)**-2H-chromen-2-one**
*(**7**): The title compound was obtained starting from **7a** and 3,3-dinitroazetidine. Analytical data for **7** (yellow solid, 43 mg, 25% yield, m.p. 98.0–99.8 °C): ESI-MS *m*/*z* 507.8 [M + H]^+^; ^1^H NMR (600 MHz, DMSO) δ 7.76 (d, *J* = 8.8 Hz, 1H), 7.64 (d, *J* = 8.0 Hz, 2H), 7.46 (d, *J* = 7.9 Hz, 2H), 7.04–6.97 (m, 2H), 4.28 (s, 4H), 4.14 (t, *J* = 4.8 Hz, 2H), 4.05 (s, 2H), 3.04 (t, *J* = 4.8 Hz, 2H), 2.45 (s, 3H). ^13^C NMR (151 MHz, DMSO) δ 160.99, 160.62, 153.25, 148.86, 144.17, 128.67, 126.61, 125.09, 120.10, 113.44, 112.43, 109.34, 100.89, 67.24, 61.10, 55.14, 31.92, 15.09.

***7-****(**3-**(**3,3-Dinitroazetidin-1-yl**)**propoxy**)**-4-methyl-3-**(**4-**(**trifluoromethyl**)**benzyl**)**-2H-chromen-2-one**
*(**8**): The title compound was obtained starting from **8a** and 3,3-dinitroazetidine. Analytical data for **8** (yellow liquid, 50 mg, 28% yield): ESI-MS *m*/*z* 521.7 [M + H]^+^; ^1^H NMR (600 MHz, DMSO) δ 7.75 (d, *J* = 8.3 Hz, 1H), 7.64 (d, *J* = 7.5 Hz, 2H), 7.46 (d, *J* = 7.5 Hz, 2H), 7.03–6.93 (m, 2H), 4.19 (s, 4H), 4.10 (t, 2H), 4.05 (s, 2H), 2.76 (t, *J* = 5.9 Hz, 2H), 2.45 (s, 3H), 1.85–1.77 (m, 2H). ^13^C NMR (151 MHz, DMSO) δ 161.02, 160.91, 153.29, 148.86, 144.18, 128.68, 126.59, 125.09, 120.01, 113.30, 112.35, 108.94, 100.83, 65.88, 60.40, 53.73, 31.91, 26.33, 15.08.

***7-****(**4-**(**3,3-Dinitroazetidin-1-yl**)**butoxy**)**-4-methyl-3-**(**4-**(**trifluoromethyl**)**benzyl**)**-2H-chromen-2-one**
*(**9**): The title compound was obtained starting from **9a** and 3,3-dinitroazetidine. Analytical data for **9** (yellow liquid, 51 mg, 28% yield): ESI-MS *m*/*z* 536.2 [M + H]^+^; ^1^H NMR (600 MHz, DMSO) δ 7.74 (d, *J* = 8.0 Hz, 1H), 7.64 (d, *J* = 6.5 Hz, 2H), 7.46 (d, *J* = 6.5 Hz, 2H), 6.98 (d, *J* = 13.9 Hz, 2H), 5.76 (s, 1H), 4.15 (s, 4H), 4.08 (t, 2H), 4.05 (s, 2H), 2.65 (t, 2H), 2.45 (s, 3H), 1.79–1.71 (m, 2H), 1.51–1.43 (m, 2H). ^13^C NMR (151 MHz, DMSO) δ 161.03, 153.31, 148.88, 144.19, 128.68, 126.56, 125.11, 119.94, 113.22, 112.41, 109.01, 100.78, 67.81, 60.35, 56.72, 31.91, 25.86, 23.00, 15.08.

***4-****((**7-**(**2-**(**3,3-Dinitroazetidin-1-yl**)**ethoxy**)**-4-methyl-2-oxo-2H-chromen-3-yl**)**methyl**) **benzonitrile**
*(**10**): The title compound was obtained starting from **10a** and 3,3-dinitroazetidine. Analytical data for **10** (yellow solid, 19 mg, 12% yield, m.p. 122.5–124.1 °C): ESI-MS *m*/*z* 464.8 [M + H]^+^; ^1^H NMR (600 MHz, DMSO) δ 7.83–7.72 (m, 3H), 7.44 (d, *J* = 7.7 Hz, 2H), 7.08–6.96 (m, 2H), 4.83 (s, 4H), 4.40 (t, *J* = 2.2 Hz, 2H), 4.33 (t, 2H), 4.04 (s, 2H), 2.44 (s, 3H). ^13^C NMR (151 MHz, DMSO) δ 160.96, 160.59, 153.24, 148.98, 145.25, 132.18, 128.98, 126.69, 119.93, 118.74, 113.56, 112.43, 108.81, 106.79, 101.06, 66.50, 63.67, 61.10, 32.22, 15.11.

***4-****((**7-**(**3-**(**3,3-Dinitroazetidin-1-yl**)**propoxy**)**-4-methyl-2-oxo-2H-chromen-3-yl**) **methyl**)**benzonitrile**
*(**11**): The title compound was obtained starting from **11a** and 3,3-dinitroazetidine. Analytical data for **11** (yellow liquid, 16 mg, 10% yield): ESI-MS *m*/*z* 478.8 [M + H]^+^; ^1^H NMR (600 MHz, DMSO) δ 7.74 (d, *J* = 8.1 Hz, 3H), 7.43 (d, *J* = 8.1 Hz, 2H), 6.97 (dd, *J* = 11.4, 2.2 Hz, 2H), 4.19 (s, 4H), 4.10 (t, *J* = 6.2 Hz, 2H), 4.04 (s, 2H), 2.76 (t, *J* = 6.9 Hz, 2H), 2.43 (s, 3H), 1.81 (p, *J* = 6.5 Hz, 2H). ^13^C NMR (151 MHz, DMSO) δ 160.99, 160.93, 153.30, 149.02, 145.29, 132.18, 128.98, 126.62, 119.72, 118.74, 113.28, 112.36, 108.94, 108.80, 100.83, 65.88, 60.40, 53.73, 32.21, 26.32, 15.09.

***4-****((**7-**(**4-**(**3,3-Dinitroazetidin-1-yl**)**butoxy**)**-4-methyl-2-oxo-2H-chromen-3-yl**)**methyl**) **benzonitrile**
*(**12**): The title compound was obtained starting from **12a** and 3,3-dinitroazetidine. Analytical data for **12** (yellow solid, 48 mg, 29% yield, m.p. 135.1–136.3 °C): ESI-MS *m*/*z* 492.8 [M + H]^+^; ^1^H NMR (600 MHz, DMSO) δ 7.74 (d, *J* = 8.1 Hz, 3H), 7.43 (d, *J* = 7.8 Hz, 2H), 6.97 (d, *J* = 11.1 Hz, 2H), 4.14 (s, 4H), 4.08 (t, *J* = 6.3 Hz, 2H), 4.04 (s, 2H), 2.65 (t, *J* = 6.9 Hz, 2H), 2.43 (s, 3H), 1.74 (dd, *J* = 13.5, 6.6 Hz, 2H), 1.52–1.43 (m, 2H). ^13^C NMR (151 MHz, DMSO) δ 161.06, 161.00, 153.32, 149.03, 145.30, 132.17, 128.97, 126.57, 119.66, 118.74, 113.20, 112.41, 109.01, 108.80, 100.77, 67.81, 60.35, 56.73, 32.21, 25.86, 23.00, 15.09.

***7-****(**2-**(**3,3-Dinitroazetidin-1-yl**)**ethoxy**)**-3-**(**4-fluorobenzyl**)**-4-methyl-2H-chromen-2-one*** (**13**): The title compound was obtained starting from **13a** and 3,3-dinitroazetidine. Analytical data for **13** (white solid, 47 mg, 30% yield, m.p. 137.9–139.2 °C): ESI-MS *m*/*z* 457.8 [M + H]^+^; ^1^H NMR (600 MHz, DMSO) δ 7.74 (d, *J* = 8.8 Hz, 1H), 7.30–7.22 (m, 2H), 7.09 (t, *J* = 8.6 Hz, 2H), 7.04–6.94 (m, 2H), 4.28 (s, 4H), 4.14 (t, *J* = 4.7 Hz, 2H), 3.93 (s, 2H), 3.04 (t, *J* = 4.7 Hz, 2H), 2.44 (s, 3H). ^13^C NMR (151 MHz, DMSO) δ 161.02, 160.51, 159.75, 153.17, 148.28, 135.25, 129.68, 129.63, 126.53, 120.89, 114.99, 114.85, 113.50, 112.37, 109.34, 100.86, 67.21, 61.09, 55.15, 31.18, 15.01.

***7-****(**3-**(**3,3-Dinitroazetidin-1-yl**)**propoxy**)**-3-**(**4-fluorobenzyl**)**-4-methyl-2H-chromen-2-one*** (**14**): The title compound was obtained starting from **14a** and 3,3-dinitroazetidine. Analytical data for **14** (yellow liquid, 16 mg, 10% yield): ESI-MS *m*/*z* 471.8 [M + H]^+^; ^1^H NMR (600 MHz, DMSO) δ 7.73 (d, *J* = 8.7 Hz, 1H), 7.28–7.24 (m, 2H), 7.08 (t, *J* = 8.7 Hz, 2H), 6.96 (d, *J* = 11.2 Hz, 2H), 4.18 (s, 4H), 4.09 (t, *J* = 6.2 Hz, 2H), 3.92 (s, 2H), 2.75 (t, *J* = 6.9 Hz, 2H), 2.43 (s, 3H), 1.80 (dt, *J* = 13.1, 6.6 Hz, 2H). ^13^C NMR (151 MHz, DMSO) δ 161.98, 161.68, 161.44, 153.85, 148.92, 135.90, 130.32, 130.27, 127.15, 121.44, 115.63, 115.49, 113.99, 112.93, 109.58, 101.44, 66.49, 61.03, 54.37, 31.81, 26.96, 15.64.

***7-****(**4-**(**3,3-Dinitroazetidin-1-yl**)**butoxy**)**-3-**(**4-fluorobenzyl**)**-4-methyl-2H-chromen-2-one**
*(**15**): The title compound was obtained starting from **15a** and 3,3-dinitroazetidine. Analytical data for **15** (white solid, 45 mg, 27% yield, m.p. 113.3–115.1 °C): ESI-MS *m*/*z* 486.2 [M + H]^+^; ^1^H NMR (600 MHz, DMSO) δ 7.73 (s, 1H), 7.26 (d, 2H), 7.09 (d, 2H), 6.97 (d, 2H), 4.14 (s, 4H), 4.08 (t, 2H), 3.93 (s, 2H), 2.65 (t, 2H), 2.43 (s, 3H), 1.81–1.67 (m, 2H), 1.55–1.40 (m, 2H). ^13^C NMR (151 MHz, DMSO) δ 161.35, 161.05, 160.93, 153.23, 148.30, 135.28, 129.68, 129.63, 126.47, 120.74, 114.99, 114.85, 113.28, 112.35, 109.01, 100.75, 67.79, 60.35, 56.72, 31.18, 25.86, 23.00, 15.00.

***3-Benzyl-7-****(**2-**(**3,3-dinitroazetidin-1-yl**)**ethoxy**)**-4-methyl-2H-chromen-2-one**
*(**16**): The title compound was obtained starting from **16a** and 3,3-dinitroazetidine. Analytical data for **16** (white solid, 31 mg, 21% yield, m.p. 173–175 °C): ESI-MS *m*/*z* 439.8 [M + H]^+^; ^1^H NMR (600 MHz, DMSO) δ 7.74 (d, *J* = 8.9 Hz, 1H), 7.27 (t, *J* = 7.5 Hz, 2H), 7.22 (d, *J* = 7.5 Hz, 2H), 7.18 (t, *J* = 7.2 Hz, 1H), 7.02–6.96 (m, 2H), 4.28 (s, 4H), 4.14 (t, *J* = 5.0 Hz, 2H), 3.95 (s, 2H), 3.04 (t, *J* = 5.0 Hz, 2H), 2.43 (s, 3H). ^13^C NMR (151 MHz, DMSO) δ 161.05, 160.48, 153.16, 148.20, 139.14, 128.26, 127.85, 126.49, 125.91, 120.96, 113.52, 112.35, 109.35, 100.86, 67.20, 61.09, 55.15, 31.97, 15.04.

***3-Benzyl-7-****(**3-**(**3,3-dinitroazetidin-1-yl**)**propoxy**)**-4-methyl-2H-chromen-2-one**
*(**17**): The title compound was obtained starting from **17a** and 3,3-dinitroazetidine. Analytical data for **17** (white solid, 34 mg, 22% yield, m.p. 115.1–116.8 °C): ESI-MS *m*/*z* 453.8 [M + H]^+^; ^1^H NMR (600 MHz, DMSO) δ 7.73 (d, *J* = 8.7 Hz, 1H), 7.27 (t, *J* = 7.5 Hz, 2H), 7.22 (d, *J* = 7.4 Hz, 2H), 7.18 (t, *J* = 7.2 Hz, 1H), 7.00–6.92 (m, 2H), 4.19 (s, 4H), 4.09 (t, *J* = 6.3 Hz, 2H), 3.95 (s, 2H), 2.76 (t, *J* = 7.0 Hz, 2H), 2.43 (s, 3H), 1.84–1.77 (m, 2H). ^13^C NMR (151 MHz, DMSO) δ 161.08, 160.76, 153.20, 148.20, 139.15, 128.26, 127.85, 126.47, 125.90, 120.87, 113.38, 112.27, 108.94, 100.80, 65.84, 60.39, 54.74, 31.96, 26.32, 13.92.

***3-Benzyl-7-****(**4-**(**3,3-dinitroazetidin-1-yl**)**butoxy**)**-4-methyl-2H-chromen-2-one**
*(**18**): The title compound was obtained starting from **18a** and 3,3-dinitroazetidine. Analytical data for **18** (yellow liquid, 22 mg, 14% yield): ESI-MS *m*/*z* 467.9 [M + H]^+^; ^1^H NMR (600 MHz, DMSO) δ 7.71 (d, *J* = 8.6 Hz, 1H), 7.26 (t, *J* = 7.3 Hz, 2H), 7.22 (d, *J* = 7.5 Hz, 2H), 7.17 (t, *J* = 7.1 Hz, 1H), 6.95 (d, *J* = 13.7 Hz, 2H), 4.14 (s, 4H), 4.07 (t, *J* = 6.3 Hz, 2H), 3.94 (s, 2H), 2.64 (t, *J* = 6.9 Hz, 2H), 2.42 (s, 3H), 1.78–1.69 (m, 2H), 1.52–1.42 (m, 2H). ^13^C NMR (151 MHz, DMSO) δ 161.09, 160.88, 153.21, 148.22, 139.16, 128.26, 127.85, 126.42, 125.90, 120.81, 113.30, 112.33, 108.94, 100.74, 67.77, 60.34, 56.69, 31.96, 25.85, 22.95, 15.02.

***7-****(**2-**(**3,3-Dinitroazetidin-1-yl**)**ethoxy**)**-4-methyl-2H-chromen-2-one**
*(**19**): The title compound was obtained starting from **19a** and 3,3-dinitroazetidine. Analytical data for **19** (white solid, 18 mg, 15% yield, m.p. 91–93 °C): ESI-MS *m*/*z* 350.1 [M + H]^+^; ^1^H NMR (600 MHz, CDCl_3_) δ 7.51 (d, *J* = 8.8 Hz, 1H), 6.83 (dd, *J* = 8.8, 2.4 Hz, 1H), 6.78 (d, *J* = 2.3 Hz, 1H), 6.15 (s, 1H), 4.29 (s, 4H), 4.13 (t, 2H), 3.09 (t, 2H), 2.40 (s, 3H). ^13^C NMR (151 MHz, CDCl_3_) δ 160.47, 154.61, 151.80, 125.14, 113.48, 111.86, 111.75, 107.95, 100.61, 67.13, 61.77, 55.60, 18.06.

***7-****(**3-**(**3,3-Dinitroazetidin-1-yl**)**propoxy**)**-4-methyl-2H-chromen-2-one*** (**20**): The title compound was obtained starting from **20a** and 3,3-dinitroazetidine. Analytical data for **20** (yellow solid, 33 mg, 27% yield, m.p. 78.5–79.6 °C): ESI-MS *m*/*z* 364.1 [M + H]^+^; ^1^H NMR (600 MHz, DMSO) δ 7.69 (t, *J* = 8.3 Hz, 1H), 6.97 (t, *J* = 9.1 Hz, 2H), 6.22 (d, *J* = 7.0 Hz, 1H), 4.37 (s, 4H), 4.10 (t, *J* = 6.2 Hz, 2H), 2.76 (t, *J* = 6.9 Hz, 2H), 2.40 (s, 3H), 1.85–1.76 (m, 2H). ^13^C NMR (151 MHz, DMSO) δ 161.42, 159.99, 154.59, 153.26, 126.32, 112.22, 110.97, 108.94, 101.06, 65.89, 60.39, 53.74, 26.32, 17.97.

***7-****(**4-**(**3,3-Dinitroazetidin-1-yl**)**butoxy**)**-4-methyl-2H-chromen-2-one**
*(**21**): The title compound was obtained starting from **21a** and 3,3-dinitroazetidine. Analytical data for **21** (yellow solid, 28 mg, 22% yield, m.p. 85.8–87.2 °C): ESI-MS *m*/*z* 378.1 [M + H]^+^; ^1^H NMR (600 MHz, DMSO) δ 7.68 (d, *J* = 8.7 Hz, 1H), 7.02–6.94 (m, 2H), 6.21 (s, 1H), 4.77 (s, 4H), 4.14 (t, 2H), 4.11 (t, 2H), 2.40 (s, 3H), 1.85–1.79 (m, 2H), 1.79–1.72 (m, 2H). ^13^C NMR (151 MHz, DMSO) δ 161.49, 160.00, 156.24, 154.60, 153.26, 126.29, 112.92, 112.27, 110.94, 106.85, 101.01, 67.66, 64.88, 24.92, 24.74, 17.96.

### 3.3. Biology

RBE (human intrahepatic cholangiocarcinoma cell lines), HOSEpiC (human ovarian surface epithelial cell lines) and T29 (immortalized but nontumorigenic ovarian epithelial cell lines) were cultured in RPMI-1640 medium (Servicebio, Wuhan, China), which was supplemented with 10% fetal bovine serum (Capricorn Scientific, Ebsdorfelgrund, Hesse, Germany) and 1% Penicillin-Streptomycin (BasalMedia, Shanghai, China). Tested compounds were dissolved into DMSO (Sigma-Aldrich, Shanghai, China) to prepare a solution with concentration of 2 mM for use.

#### 3.3.1. In Vitro Anti-Proliferative Assay

The in vitro antiproliferation of the chemical compounds was measured by the MTT reagent, as described in the literature. Briefly, 4000–6000 cells in 100 μL of medium per well were plated in 96-well plates. After incubated for 24 h at 37 °C, the cells were treated with different concentration of tested compound or DMSO (as negative control) for 48 h. At the same time, blank group without adding cells was set. Then, the medium per well was replaced with 150 μL of fresh medium containing 10% MTT (5 mg/mL in PBS) in each well and incubated at 37 °C for 4 h. Last, the MTT-containing medium was discarded and 150 μL of DMSO per well was added to dissolve the formazan crystals newly formed. Absorbance of each well was determined by a microplate reader (Synergy H4, Bio-Tek) at a 570 nm wavelength. The inhibition rates of proliferation were calculated with the following equation:Inhibition ratio (%) = (OD_DMSO_−OD_compd_)/(OD_DMSO_−OD_blank_) × 100 

The concentrations of the compounds that inhibited cell growth by 50% (IC_50_) were calculated using GraphPad Prism, version 6.0.

#### 3.3.2. Measurement of Intracellular NO

Intracellular NO was measured with 3-amino,4-aminomethyl-2′,7′-diflfluorescein, diacetate (DAF-FM DA, Beyotime, Shanghai, China). In detail, RBE cells in the logarithmic growth phase were collected and spread on 6-well plate at a density of 150,000 per well overnight. The adherent cells were pretreated with 5 μM DAF-FM DA at 37 °C for 20 min and then incubated with RRx-001, compounds **2–6** and **21** for 2.5 h, followed by flow cytometer analysis (BD Accuri C6, Shanghai, China). Cells were washed three times with cold PBS between each step. The experiment was performed three times.

#### 3.3.3. Cell Cycle Analysis

The experimental cells RBE were cultured and collected when the cells were in good growth state. After digesting with 0.25% trypsin (Beyotime), the cells were collected and centrifuged. The cells were inoculated in petri dishes (Φ = 6 cm) with an inoculation density of 500,000 cells per dish and 3 mL medium per dish. The cells were placed in the incubator, changed to serum-free RPMI 1640 medium after 12 h, left to continue to culture for 12 h, and then compounds were added. Compounds groups with the concentration of 1 μM and control group treated with DMSO were set. The original culture medium was removed, and the compounds were added. After incubation for 16h, the cells were collected and centrifuged at 1000 r for 5 min. The cells were rinsed twice with PBS, and the supernatant was removed by centrifugation. 75% ethanol (1 mL) was added and placed in a refrigerator at −20 °C overnight for cell fixation.

The fixed cells were centrifuged at 2000 r for 5 min, the supernatant was discarded, 1 mL of PBS solution was added, and the supernatant was resuspended and discarded. 0.5 mL of PI staining solution was added to each tube (staining buffer: 20 × PI staining solution: 50 × RNase A = 100:5:2) and stored at room temperature for 30 min in the dark for testing on the flow cytometry.

#### 3.3.4. Western Blot Analysis

Approximately 2 × 10^5^ cells of RBE were seeded in 6-well plates. After treated with compound **3** (1 and 4 μM) for 24 h, cells were lysed with ice-cold RIPA lysis buffer (Beyotime, Shanghai, China) for 30 min and then centrifuged at 12,000 rpm for 10 min at 4 °C. The total protein concentration was determined by BCA protein assay kit (Beyotime, Shanghai, China). Equal amounts (20 μg per load) of protein samples were subjected to 12% SDS-PAGE gel and transferred onto polyvinylidene fluoride (PVDF) membranes (Epizyme Biotech, Shanghai, China), which were then blocked with 5% bovine serum albumin (Ebsdorfelgrund, Hesse, Germany) for 1.5 h and reacted with primary antibodies (1: 1000 diluted) at 4 °C overnight. Subsequently, the PVDF membranes were incubated in the secondary antibody (1: 10,000 diluted) for 2 h at room temperature. ECL chemiluminescent solution was used for color rendering. The antibodies against Cyclin B_1_, Caspase 3, and PARP were purchased from Cell Signaling Technology. The secondary antibodies conjugated with horseradish peroxidase (HRP) were from AoWei Biology and CST. The protein bands were developed by the chemiluminescent reagents (Meilunbio, Shanghai, China).

#### 3.3.5. Metabolic Stability in Liver Microsomes

Microsomes in 0.1 M TRIS buffer pH 7.4 (final concentration 0.33 mg/mL), cofactor MgCl_2_ (final concentration 5 mM), the tested compound (final concentration 0.1 μM, co-solvent (0.01% DMSO), and 0.005% Bovin serum albumin (BSA)) were incubated at 37 °C for 10 min. The reaction was started by the addition of NADPH (final concentration 1 mM). Aliquots were sampled at 0, 7, 17, 30, and 60 min, respectively, and methanol (cold in 4 °C) was added to terminate the reaction. After centrifugation (4000 rpm, 5 min), samples were then analyzed by LC-MS/MS.

## 4. Conclusions

In this study, twenty novel NO donor compounds **2–21** were synthesized by coupling dinitroazetidine moiety and coumarin scaffold with amide and aliphatic carbon chain linkers, respectively. The antiproliferation activity evaluation of them showed that five compounds **2–5** and **21** had a strong inhibition activity in human intrahepatic cholangiocarcinoma cell lines RBE comparable with RRx-001 and displayed weak cytotoxicity to two normal cell lines HOSEpiC and T29. These five hybrids and RRx-001 all could release effective concentrations of NO in RBE cell lines, which supposed that high anticancer potency of these NO donors was positively associated with their intracellular NO release levels. The preliminary mechanism research revealed that compound **3** could arrest RBE cells cycle at G_2_/M phase and apparently downregulate the expressions of cell-cycle- and apoptosis-related proteins Cyclin B_1_, Caspase-3, and PARP. Moreover, compared to furoxan-coumarin hybrid **CY-14S-4A83**, dinitroazetidine-coumarin compound **3** showed an obviously improved metabolic stability in the tested liver microsomes. Overall, compound **3** was deserved further to study for developing an ideal lead compound with anticancer activity.

## Data Availability

Not applicable.

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
