# Peer review of "Novel Nitric Oxide Donor Dinitroazetidine-Coumarin Hybrids as Potent Anti-Intrahepatic Cholangiocarcinoma Agents"

_molecules, 2022, doi:10.3390/molecules27134021_

Round 1

Reviewer 1 Report

The author did choose interesting molecule of great interest for scientific community. The overall this research article adds information of anticancer of nitric oxide(NO)donors, Dinitroazetidine-Coumarin Hybrids. The paper needs to address focus point of using Coumarin rather than any other class of compounds. Some of areas the review missed include;

Authors have reported that these potent derivatives compounds kill cancer while have low cytotoxicity toward selected normal cells. This data should be supported with two in vivo mouse model data atleast for compound 3.

Reviewer 2 Report

The manuscript represented interesting data and is rather well written. Chemistry is fine, however, there are some points to be improved in the biological part of the study.

  • Table 1: Please determine the IC50s of the compounds in all the cell lines and please calculate the selectivity index.
  • Figure 3: Please mark the positive cells and show the numbers
  • Figure 4a: Please show the numbers and/or statistical significance of the difference.
  • Figure 4b: Please quantify the band intensity and normalize it to the loading control. Cyclin 1B and beta-actin go down somewhat simultaneously. Please also show CLEAVED caspase and PARP.

Allover the manuscript there are some unexplained abbreviations which are for example MFI, MF, etc. Please explain them and the others.

Round 2

Reviewer 1 Report

Authors claims that they may published invivo data in upcoming study. Paper can be accepted.

Author Response

Thank you very much for your affirmation and recognition of the manuscript.

Reviewer 2 Report

The authors have made some of the requested changes. However, the manuscript still can be improved, especially with regard to cleaved-caspase and PARP Western blotting analysis. At the same time, the authors have mentioned limited access to the lab due to the current COVID-19 restrictions in China. Therefore, I believe that the editor should make a final decision about this manuscript.

Author Response

Thank you very much for your evaluation and comments. And in Figure 4b, we supplemented the loading control figure of Cyclin B1, Caspase-3 and PARP compared to beta-Actin. However, it is very pity that we still have no chance to finish cleaved-caspase and PARP Western blotting analysis right now.